# Strategies to Improve Neonatal Intubation Safety by Preventing Endobronchial Placement of the Tracheal Tube—Literature Review and Experience at a Tertiary Center

**DOI:** 10.3390/children10020361

**Published:** 2023-02-11

**Authors:** Joaquim M. B. Pinheiro, Upender K. Munshi, Rehman Chowdhry

**Affiliations:** Division of Neonatology, Department of Pediatrics, Albany Medical College, Albany, NY 12208, USA

**Keywords:** infant, neonate, intubation, endotracheal tube, airway, safety, adverse events, quality improvement

## Abstract

Unintended endobronchial placement is a common complication of neonatal tracheal intubation and a threat to patient safety, but it has received little attention towards decreasing its incidence and mitigating associated harms. We report on the key aspects of a long-term project in which we applied principles of patient safety to design and implement safeguards and establish a safety culture, aiming to decrease the rate of deep intubation (beyond T3) in neonates to <10%. Results from 5745 consecutive intubations revealed a 47% incidence of deep tube placement at baseline, which decreased to 10–15% after initial interventions and remained in the 9–20% range for the past 15 years; concurrently, rates of deep intubation at referring institutions have remained high. Root cause analyses revealed multiple contributing factors, so countermeasures specifically aimed at improving intubation safety should be applied before, during, and immediately after tube insertion. Extensive literature review, concordant with our experience, suggests that pre-specifying the expected tube depth before intubation is the most effective and simple intervention, although further research is needed to establish accurate and accepted standards for estimating the expected depth. Presently, team training on intubation safety, plus possible technological advances, offer additional options for safer neonatal intubations.

## 1. Introduction

Endotracheal intubation is an essential procedure in critical care, allowing life-supporting assisted ventilation and providing a conduit through which drugs and devices can reach the lower airways, but it has significant risks whether attempts fail or succeed. Intubation success, generally defined as placing the endotracheal tube (ETT) within the trachea, requires specialized training and the experience of approximately 40 intubations to achieve proficiency [1]. Failed intubation attempts are common [2], particularly in neonatal care, where difficult airways are frequent [3,4]. Patient safety can be compromised during tracheal intubation through failed or prolonged attempts that cause physiologic destabilization, but also through physical trauma, procedure-induced pain and stress, and malposition of the ETT within the tracheal airway, as is the case with deep tracheal or endobronchial intubation [5,6,7,8,9,10].

Inadvertent right mainstem intubation or similar deep malposition of the ETT was reported in earlier studies in up to 58% of neonates [10,11,12]. In a more recent prospective study on complications of neonatal intubation, the NEAR4NEOS collaborative reported that mainstem intubation was diagnosed by X-ray in 2% of NICU intubations [13], which is a lower rate than other reports. Using similar methods, the NEAR4KIDS collaborative, reported mainstem intubation in 13% of PICU patients [14]. In our setting, ETT tip beyond the T4 vertebral level was observed in 10.5% of newborns [15].

Sequelae of endobronchial intubation include ineffective ventilation, which is associated with intensive resuscitation in the delivery room [16], atelectasis [11,17], pneumothorax [18], other major morbidities [17], and death [19]. We have observed these and other sequelae including bronchial mucosal trauma, which may result in granuloma formation and subsequent obstruction. [Figure 1] Tube thoracostomy for asymmetric breath sounds has been reported as a complication of endobronchial intubation in critically ill children, in whom the incidence of bronchial intubation may be as high as 26% [20]. Nevertheless, trigger tool methods to identify adverse events in NICUs did not specifically target endobronchial intubations [21], and the need to include these under the scope of iatrogenic harm has been subsequently highlighted [22].

Estimates of the burden of morbidity attributable to endobronchial intubation are lacking. In an audit of safety incident in a UK NICU, despite a 54% incidence of deeply positioned ETTs, these were typically not reported to the National Patient Safety Agency [12]. The paucity of such adverse event reports in the US was corroborated through our participation in a multicenter initiative on NICU safety improvement [15,23]. Although the sequelae of endobronchial intubation have not been quantified, a survey revealed that more than one third of academic neonatologists believed that it may contribute to neonatal morbidities at least as much as medication errors; however, the majority of their NICUs have not monitored this problem through quality assurance [15].

Given the evidence that endobronchial intubation is underreported, and our own observations that the complications of neonatal intubation are rarely attributed to deeply positioned ETTs, even when this appears likely, we undertook a long-term project to improve the accuracy of initial ETT placement and minimize the incidence of endobronchial intubation, thereby decreasing the risk of associated complications. In this report, we describe our approach to developing and implementing strategies to decrease endobronchial intubation in neonates at our tertiary center.

## 2. Materials and Methods

### 2.1. Clinical Environment and Patient Population

The 60-bed NICU at Albany Medical Center (AMC) is a Level IV regional perinatal unit with approximately 800 annual admissions, of which about one third are outborn at referring hospitals in a 25-county region. Neonatal intubations by our teams are performed by neonatology attending physicians or fellows, residents, as well as respiratory therapists or advanced practice providers, who jointly staff the neonatal transport team. Intubations performed pre-transport by clinicians from referring hospitals were qualified as outside intubations.

### 2.2. Quantification of Endobronchial Intubation Risk

Early in this project, as X-ray technology was transitioning from film to digital media, we realized that despite image enhancement, the carina could not always be identified with certainty by independent observers; furthermore, the position of the carina was variable, and it could be above the 3rd thoracic vertebra level (T3) [24]. Since we aimed for an approximately mid-tracheal location of the ETT tip, between the upper border of T1 and the lower border of T2 [25], ETTs positioned outside of this range would most typically be adjusted; those whose tip lay beyond the body of T3 must be adjusted, as they were too “deep”, whether ending in a mainstem bronchus or just above the carina. Therefore, to characterize the safety of the intubation process in a rigorously measurable manner, we specified the method for identifying vertebral levels reproducibly [24], then operationally defined a “deep ETT” (dETT) as that whose tip was beyond the body of T3. Because this is not an exact correlate of endobronchial intubation, clinicians caring for neonates whose ETT is in the T1-T2 target range, but whose carina is relatively high, will adjust the ETT position so that the tip will lie at least 1 cm above the carina. If an ETT was initially secured but then adjusted prior to the first X-ray, only the position during the X-ray was considered in the analyses.

### 2.3. Characterization of Deep ETT Position at Baseline

In the Summer of 2000, we reviewed the initial chest X-rays pertaining to the 32 most recent consecutive neonatal intubations. The proportion of dETTs (beyond T3) was 47%, and an additional 13% were between T2 and T3 with 6% above T1; thus, 34% were in the target range. We aimed to decrease the proportion of dETTs to 10% or less within 1 year, reasoning that this should result in a minimal incidence of endobronchial intubations. To this end, we trended the proportion of dETTs for each subsequent month using control charts; monthly proportions on these p-charts were considered to violate statistical control rules if they were outside the 3-sigma control limits (IBM SPSS Statistics v. 28, Armonk, NY, USA). We reported results at monthly Departmental Mortality and Morbidity/QI conferences, incorporated them into the NICU quality and safety performance dashboard, and posted them as a simple graph to provide feedback for all staff. These activities were reviewed and approved by the Albany Medical Center Institutional Review Board as quality improvement work exempt of the need for informed consent (protocol #3937).

### 2.4. Initial Intervention: Ensuring That Expected ETT Depth Was Pre-Specified

Simple observation of actual neonatal intubations revealed that expected ETT depth was rarely pre-specified explicitly, although clinicians assumed they were doing this. In reality, the verification of ETT depth was carried out only after ETT insertion during the assessment of response to ventilation, as noted in the Neonatal Resuscitation Program flow diagram [26]. We developed an Ishikawa (fishbone) diagram to explore the contributing causes of deep intubation [27]. We noted that the documentation of ETT depth in the respiratory care order, procedure note, and respiratory therapist notes was inconsistent, as it did not follow a set format. This structure neither forced nor prompted the clinical team to be mindful of ETT depth prior to initiating the intubation. Borrowing from the concept of hazard analysis commonly applied to food safety (HACCP) [28], we considered the expected ETT insertion depth to be a critical control point in intubation safety. Consequently, we redesigned the neonatal intubation flow diagram (Figure 2) to reflect the importance of its prospective estimation, as well as its verification by both the primary proceduralist and the key assistant. This diagram was used as the basis for staff education, emphasizing the notion that intubation safety is the responsibility of all team members involved in the procedure.

### 2.5. Determining and Applying the Expected ETT Depth

Expected ETT depth was initially obtained by applying Tochen’s formula (ETT depth in cm = weight in kg + 6) [29] to body weight or, for delivery room intubations, expected weight. This was incorporated in the bedside emergency medications “code sheet” and amended, if necessary, after the corrected, adjusted depth was determined from the post-intubation X-ray. If reintubation was necessary, the recent corrected ETT depth was used. Beyond staff education, structural changes that helped consolidate awareness of ETT depth were gradually implemented over the next 4–5 years, including the redesign of procedure notes, delivery room resuscitation sheets, respiratory therapy documentation sheets, bedside airway cards and respiratory care orders to specify ETT depth. We also attempted to implement marking the expected depth on the ETT using a red or black ink marker, but there was low compliance with this practice.

Variation in the measurement of ETT depth was subsequently minimized by standardizing the anatomical reference point as the mid-upper lip. Retrospective analyses of ETT depth records that had been verified through bedside airway audits allowed for the refinement of the expected ETT depth algorithm in our code sheets, as it became obvious that Tochen’s formula [29] was significantly inaccurate in extremely low birthweight neonates [30], in whom our estimates aligned closely with those of the original NICU Tools algorithm [31]. Further analyses (J. Pinheiro, unpublished) revealed that in large-for-gestational age newborns, weight-based ETT depth tends to overestimate appropriate mid-tracheal depth; thus, since 2009, the code sheet has alerted users to this fact, and prompted alternate use of the nasal-tragus length measurement to pre-estimate ETT depth [32], which has more recently been recommended by the Neonatal Resuscitation Program [33].

## 3. Results

### 3.1. Factors Contributing to Deep Intubation

Review of multiple instances of deep intubation in the early stages of the project, along with brainstorming by a multidisciplinary team, elicited multiple observed and perceived causes underlying deep intubation. These were distributed among the categories of Materials, People, Policies/Guidelines and Methods/Procedures, and are detailed in Figure 3.

### 3.2. Deep Intubation Rates during Implementation

From the start of the project through November 2022, a total of 5754 initial intubations have been reviewed; of these, 5237 were performed by our AMC NICU team, and 517 were carried out by clinicians at the referring hospitals, before transport to AMC. The proportion of dETT was 47% at baseline and it decreased with the initial interventions, so that no subsequent points were out of statistical control (Figure 4). A shift towards lower dETT rates occurred after the first 10 months, and the 18% overall rate did not change significantly after that time. This is more obviously apparent in Figure 5, in which the results are aggregated by project year.

### 3.3. Proportion of Deep ETT According to Intubation Location and Staff

Although we wished to use similar safety standards in both inborn neonates and those born at referring hospitals, the key aspects of our initial intervention were not immediately applicable at referring hospitals. Therefore, we partitioned the intubation data by location and intubating team: intubations performed by our team, whether at AMC or on transport, were categorized as “inborn”, whereas those performed by staff from referring hospitals, prior to transport, were considered “outborn” intubations. As shown in Figure 5 and noted above, the proportion of inborn intubations with dETT decreased rapidly, whereas dETT in outborn intubations decreased gradually over two decades. Through the years, the rates of dETT in outborn intubations have remained significantly higher than in the inborn group, despite attempts to extend the principles of intubation safety to referring hospitals through regional perinatal education and staff migration.

Notable in Figure 5, our initial goal of achieving dETT rates <10% was attained in only three consecutive years, 2015–2017, which followed improvement after an apparent spike in 2012. No specific data on staffing are available, but this period was correlated with a significant influx of new clinicians, which prompted extended re-education efforts on our processes. The most recent five years have also witnessed turnover of most of the frontline nursing and respiratory therapy staff; there is limited ability to address details of the intubation procedure during onboarding, so modules on neonatal airway management are presented to staff at the annual Skills Fair and reinforced by monthly feedback of dETT data.

## 4. Discussion

### 4.1. Interpreting Our Observations

Our report summarizes long-term efforts to minimize endobronchial intubations in a regional NICU where initial improvements were noted but not sustained towards reaching desired goals; concurrently, deep endotracheal tube insertion continues to occur at a higher rate in neonates intubated at referring hospitals. Although we have relied mostly on quality improvement (QI) methodologies, the work entered a long maintenance phase during which there has been nearly complete and repeated turnover of clinical staff. Staff inexperience, unfamiliarity with our methods, plus the constant introduction of new trainees in our teaching environment, would be expected to drive dETT rates towards pre-intervention rates. We suspect that the improved and relatively stable rates of deep intubation reported here are due to the safety processes we implemented, which likely helped prevent the degradation of intubation performance, while being insufficient to achieve our original target.

### 4.2. Comparison with Related Studies

The relative ease with which we obtained initial improvement in dETT rates using simple interventions and minimal resources parallels the report of Subhedar et al. [12], in which a clinical audit and introduction of a pocket sized “ready-reckoner” for ETT depth prior to intubations resulted in a decrease in ETT malposition from 54% to 32%. In our setting, the new focus on the dETT problem and the addition of an ETT depth calculator had a similar impact. A more recent neonatal intubation QI project used as an intervention a pre-procedural “Intubation Timeout” list which included expected ETT depth based on Tochen’s formula [34]. This may be an important feature compared with processes that only force explicit verification of ETT depth after its insertion [35,36].

### 4.3. Sequelae of Deeply Placed ETTs

Despite the high frequency of potentially preventable ETT positioning errors, the cases in Subhedar’s audit were not specifically reported to the National Patient Safety Agency, and even if some were included under procedure-related incidents, such reports must have been infrequent [12]. Similarly, in our setting, dETT events were not formally reported, even when retrospectively identified and possibly associated with morbidities; lack of dETT event reporting was also noted among centers participating in a NICU safety improvement collaborative [15,23]. Routine reporting of dETT as a “plastic dosing” medical error would require a substantial time commitment due to the frequency of these events; furthermore, possible sequelae of endobronchial intubation may be difficult to attribute to ETT malposition, thus precluding the estimation of sequelae of endobronchial intubation. For example, when tracheal intubation is followed by a poor response to ventilation, including severe bradycardia or signs suggesting a pneumothorax, neonatal intubation algorithms prompt clinicians to assess ETT position and adjust to an appropriate depth [35], so that only the corrected position is documented on subsequent imaging. This scenario is particularly likely in delivery room resuscitations, where intubation often occurs without a baseline period of cardiorespiratory stability. In such case, a severe tracheal intubation associated event (TIAE) would not be attributable to a transient, undiagnosed endobronchial intubation in the NEAR4NEOS classification; in this system, an otherwise uncomplicated mainstem intubation, noted on postintubation chest x-ray, is classified as a non-severe TIAE [37].

We did not attempt to collect information on sequelae associated with dETT because considerable resources would have been required. Furthermore, we hoped to characterize the frequency of dETT as an obvious threat to patient safety, leveraging those data to sensitize and encourage staff to engage in preventing dETT without the stigma of specific morbidities linked to individual events. We presumed that by decreasing the dETT rates so markedly, we concurrently decreased associated pulmonary sequelae, as reported by other authors; [38] of note, ETT-associated complications, such as uneven lung expansion, decreased significantly by targeting ETT tip alignment with T1-T2 based on a gestational age-guideline, even without the ETT being specifically in an endobronchial location [25].

### 4.4. Methods for Preventing Endobronchial Intubations

Notwithstanding the difficulty in estimating the burden of morbidity associated with endobronchial intubation, its prevention is an implicit goal in neonatology practice guidelines; dETT positioning is rarely indicated [39,40]. Therefore, we conducted a further literature review aiming to identify approaches to preventing dETT in neonates. As is obvious from Figure 2, interventions to prevent dETT can be applied at multiple stages of intubation, using redundant safety checks. This principle has been substantiated in an RCT in anesthetized adults, showing that explicit targeting of ETT depth was the most useful single intervention to prevent endobronchial intubation, but the combination of ETT depth, lung auscultation, and visualization of symmetrical chest movement was necessary to achieve optimal sensitivity and specificity [41]. Accordingly, we consider potential interventions to prevent endobronchial intubation as those applicable before, during and after ETT insertion, with the understanding that a practical combination of interventions will be needed for optimal safety and that these interventions must be integrated with others aimed at maximizing intubation success rates and minimizing its complications and harms [42,43].

#### 4.4.1. Declaring Predicted ETT Depth during Preparation for Intubation

The phase of preparation for ETT insertion provides the best opportunity to prevent dETT. To ensure that this crucial step actually occurs and is shared by the entire intubating team, it should be integrated as a mandatory element in the preparatory phase of intubation. This can be achieved by specifying expected ETT depth (and possibly size) in orders for elective intubations, including ETT depth on a bedside code card used for emergency intubations, and verbally specifying this depth as part of the pre-procedure time-out; compliance with the latter may be prompted by listing it in the pre-intubation checklist [34]. In the delivery room, a table on the wall [35] or a pocket card [12,44] can be used. Determining the expected ETT depth accurately is crucial but not simple, as studies have yielded inconsistent results. Tracheal length varies significantly over the range of sizes of intubated neonates, although the distances between the mid-trachea and the carina or the glottis may be as small as 1–2 cm [24], resulting in very narrow tolerances for the ETT movement that inevitably occurs during clinical care [45]. Tochen’s formula was a practical and effective early predictor of ETT depth [29], but due to the non-linearity of the weight-tracheal length relationship [24], it promotes dETT placement in ELBW neonates [25,30]; furthermore, it is not strictly applicable to newborns requiring intubation in the delivery room before being weighed. Therefore, the NRP currently recommends using a combined weight and gestational age table to predict ETT depth at the lip [35]; alternatively, it suggests calculating the expected lip-to-tip distance for orotracheal intubations from (nasal-tragus length + 1) cm [32]. However, a recent prospective evaluation of this approach has shown that both nasal-tragus length and Tochen’s methods are more reliable than the weight or gestational age-based tables in the NRP, with the latter tables yielding rates of optimal ETT positioning below 20% [46]. It is unclear whether such discrepant observations result from using different external anatomical reference points for ETT depth, such as the lip [35] versus the gum [34], failure to consistently position the head during post-intubation chest X-rays [45], lack of standardization in counting the vertebral level abutting the ETT tip [24], or any combination of these factors and other unreported variables, such as infant weight relative to gestational age. A systematic review and meta-analysis of methods to estimate ETT insertion depth revealed that the most commonly used methods are inaccurate and unreliable, typically resulting in malposition of about half of neonatal ETTs [47]. The current uncertainties about the optimal method of pre-estimating accurate ETT insertion depth have prompted an ongoing randomized trial comparing gestational age and nasal-tragus length [48]. Meanwhile, the algorithm used for predicting ETT depth in our NICU has evolved over the years, with refinements following introduction of specifications such as mid-upper lip for standard anatomical reference, changes in ETT holding devices, etc. We suggest that centers trying to implement any method for predicting ETT depth should verify its accuracy in their setting by applying strict measurement methods and systematically auditing predicted and adjusted ETT depths in a sample of at least 30–50 neonates comprising a full range of body sizes. To this end, having a form with prompts for documenting both post-intubation and final (post-adjustment) ETT depth by the intubating clinician as well as the respiratory therapist will help promote awareness of dETT as part of safety culture, while also serving to facilitate quality and safety audits.

#### 4.4.2. Inserting the ETT to the Desired Depth

##### Glottic Depth Markings

The stage of actual ETT insertion presents further opportunities to prevent deep intubation prior to providing assisted ventilation. The most widely available method relies on using markings near the ETT tip that are intended to be aligned with the vocal cords, which should place the tip of an appropriate size ETT near the mid-tracheal level. Historically, these glottic depth markings were the first safety feature designed into neonatal tracheal tubes aimed specifically at preventing endobronchial intubation. It was a single transverse black line near the patient end of the ETT, positioned at 22, 24, and 26 mm from the tip of 2.5, 3.0 and 3.5 nominal size ETTs, respectively; this would place the ETT tip near the mid trachea when the tube is used in neonates whose weights are in the appropriate range for that ETT size [49]. Although those initial markings were effective in decreasing complications of endobronchial intubation, their effectiveness was not reevaluated in the next two decades as increasingly small neonates were routinely intubated. In addition, wide variation in the type and position of these markings among ETT manufacturers has resulted in confusion among clinicians that leads to tube malposition [50,51,52,53]. Furthermore, some manufacturers added a second or even third set of lines proximal to the distal marking, without clear instructions for use, which resulted in variable understanding and usage of this safety feature, even among expert intubators of neonates [15]. If the second or third set of lines, generally located 1 and 2 cm proximal to the distal line, are apposed to the vocal cords of a neonate, the distal tip of the ETT will usually lie in the distal trachea or mainstem bronchus. Recent work on the standardization of the nomenclature as well as requirement for including the distances between the glottic depth markings and the ETT tip on the individual unit packages provides clinicians with specific information on the device features at the point of care, but evidence-based guidelines on the use of each set of markings during intubation are still lacking [50,54], An RCT comparing insertion guided by the glottic depth mark versus Tochen’s formula showed that about half of the ETTs were too deep in both groups [55]. The unsatisfactory performance of currently available markings in ensuring safe ETT placement makes the use of this feature particularly inappropriate for ELBW neonates. Recently published data on tracheal dimensions obtained from measurements at autopsy or digital imaging in neonates of varying sizes, including 22 weeks’ gestation, can be used to inform the redesign of glottic depth markings so that they can contribute to safer intubation of the smallest, highest-risk neonates [24,56,57]. Redesigned neonatal ETTs may help reduce harm events related to inappropriate ETT depth [22]. More appropriately positioned markings, along with their consistent use among clinicians, should result in more accurate placement of the ETT tip in the mid-trachea. Nevertheless, a disadvantage of this method is that using direct laryngoscopy, only the intubator can visualize the position of the glottic depth marking relative to the cords, which precludes a safety verification by an assistant or supervisor. Furthermore, with removal of the laryngoscope, the stylet, and slight manipulation of the tube while it is being secured, the position of the glottic depth marking can change inadvertently. Therefore, although the glottic depth marking remains a useful feature that should be improved further, we suggest that it should be used as an initial safety stop for insertion by the intubator, to be followed by verification of ETT depth at the lip by both the intubator and assistant; as noted in Figure 2, if the ETT depth deviates substantially from the predicted depth, it should be adjusted before assisted ventilation is applied since there is significant risk in ventilating a single, non-compliant lung, and auscultation of breath sounds is widely known to be an unreliable indicator of appropriate ETT position [33].

##### Videolaryngoscopy

The use of videolaryngoscopy has been associated with a decrease in tracheal intubation-associated events [58], but it has not been studied as an aid to preventing dETT. It is conceivable that in conjunction with appropriately designed glottic depth markings, this technique could allow an intubation assistant or supervisor to visually verify the appropriate position of the marking, particularly when intubations are performed by trainees [59].

##### Other Aids for Initial ETT Positioning

Digital suprasternal palpation of the ETT tip is a simple technique to detect the ETT tip near the mid-trachea [45]; in an RCT, this method yielded results similar to Tochen’s formula, both achieving satisfactory ETT position in only about half the subjects [60]. Additional technologies that have been used to assist in positioning the ETT near the mid-trachea before applying assisted ventilation include fiberoptic bronchoscopy [61], which is impractical for routine use. Potentially usable and apparently effective devices such as a light wand incorporated in the ETT that allows transthoracic visualization of the tip region [62] and a metal ring on the ETT that can be located using an external magnetometer [63], were never adopted beyond their initial trials. Finally, point-of-care ultrasound is increasingly available, and it can accurately assess ETT position [64,65]. To date, it has been used mostly after ventilation and ETT securement; it remains unclear if it can be applied at this stage of the intubation, where decisions must be made in a few seconds.

#### 4.4.3. Verification of ETT Position

In the final stage of intubation, when assisted breaths are provided through the ETT, tube malposition remains an immediate concern. Although consistent detection of exhaled CO_2_ has become a standard of care to rule out esophageal intubation [35,66], it fails to detect bronchial intubation in adults under anesthesia [67], and this has not been studied in newborns. Clinical evaluation, including auscultation of breath sounds, is also unreliable in detecting endobronchial intubation in infants and small children [68], and should not be used alone to assess ETT position [35]. A new technology that can rapidly detect both esophageal ventilation and asymmetric lung ventilation within a few breaths is electrical impedance tomography [69,70], however, the detector must be removed while obtaining radiographs and it may limit traditional auscultation, so its practical application during neonatal intubations remains to be developed.

##### Post-Intubation X-ray and Ultrasound Imaging of ETT

Due to the limitations of the above methods in ensuring the accurate positioning of the ETT and preventing endobronchial intubations, post-intubation chest X-ray remains a standard in neonatal practice, with the possible exception of intubations performed in the operating room. Aside from radiation exposure and the time delay in the availability of the images, this method has other shortcomings including occasional difficulty in visualizing the carina and the need for methodical counting of vertebral bodies to ensure precise adjustments of ETT depth [24]. Significant ETT migration with changes in head position (1.2 cm cephalad with head turning and 0.5 cm caudad with neck flexion) may lead to inappropriate adjustments when the relative position of the head is not standardized or considered [45]; this problem is compounded by 1–2 cm excursion of the carina during respiration, while synchronization of X-ray exposure with the respiratory cycle of neonates is generally impossible [45]. Advances in image analysis and artificial intelligence may help overcome the problem of non-standard head positions in neonatal X-rays, with a mathematical algorithm suggesting accurate corrections of ETT position, based on analysis of CXR features including ETT, head and chest landmarks [71].

#### 4.4.4. ETT Securement and Dynamic Displacement

Ensuring that the ETT position remains appropriate after the ETT is initially secured and adjusted remains an important task, to prevent subsequent endobronchial migration of the ETT as well as unplanned extubations, either of which can result from ETT migration due to head and/or holding device movement, slippage through the holder, and inappropriate adjustments after subsequent X-rays. Change in tracheal length in chronically intubated neonates is a minor factor, increasing at about 1.3 mm per week [24]. Monitoring of ETT depth during critical care, along with vital signs and its documentation by nurses and respiratory therapists, is essential to detect ETT migration from its initial position. When subsequent imaging suggests malposition, it is important to trend the ETT depth using the electronic record or bedside airway card to verify that this was not a transient displacement before making potentially unnecessary and risky adjustments to ETT position. In this setting, point-of-care ultrasound may be particularly useful to double-check the location of the ETT tip without repeating an X-ray [64]. The recent availability of an ultrasound-based sensor of ETT position relative to the carina may aid in detecting tube migration in real time, thus allowing closer control of its position as well as avoidance of inappropriate ETT adjustments following transient displacement during X-rays [72].

### 4.5. Limitations

Our study and the published literature have significant limitations as guides for practice. Although we used primarily methods from quality improvement and patient safety work, implementation of the interventions began even as we were retrospectively determining the baseline incidence of dETT through review of recent X-ray films, prompted by recent cases with complications of intubation. This limited our ability to characterize the baseline variation in intubation performance, which would have required several months of pre-intervention data. The intensity of the interventions diminished after the initial interventions due to limited resources; therefore, the potential for achieving further performance benchmarks in preventing deep intubation of neonates remains unknown. We suspect that the secular decrease in dETT rates in neonates intubated by staff from referring hospitals is not likely due to outreach education, but it may reflect increasing pre-intubation advice telephone from our own clinicians, the introduction of a cell to document ETT depth in our transport communication forms, and the gradual spread of our own trainees, former respiratory therapists and neonatal advanced practice providers through the region. The generalized application of our methods in other settings would require a significant time commitment which may make it feasible for only brief periods, particularly by partnering with pediatric radiologists, but this may be facilitated in the foreseeable future by using artificial intelligence to automate the accurate detection of ETT position [71]. Finally, because our project was not designed to collect data on dETT sequelae, we cannot quantify the value of these efforts in terms of harm avoided. The existing literature on dETT provides us with limited evidence-based best practices to minimize this problem, aside from abandoning Tochen’s formula as the primary estimator of ETT depth [35]. However, it points towards potential advances in neonatal ETT design and technologies for improved detection of ETT position. Furthermore, both published work and our experience reported herein suggest that significant improvements in dETT rates can be attained quickly through simple awareness and implementation of basic safety principles. As a final limitation, neither published studies nor our own tracked unplanned extubation rates as a balancing measure when aiming to minimize dETT rates.

## 5. Conclusions

In sum, we documented the paucity of research and QI work focusing on neonatal endobronchial intubation as a patient safety problem, the relative ease of attaining substantial improvements initial using simple safety principles and interventions at various stages of the procedure, as well obstacles to minimizing dETT rates in the long-term. We speculate that in order to rapidly implement and effectively sustain processes to prevent dETT, recurrent staff education, data feedback, and integration of ETT depth awareness in practice tools, must be supplemented by team training, particularly when the team includes trainees [73]. Whereas insertion of the ETT in the trachea is principally dependent on the intubator’s experience and motor skills [74,75], adjusting the ETT to the appropriate depth before initiating assisted ventilation can be assigned to two or more team members. This task can be practiced in a simulation environment designed to enhance team skills, as team performance measures have been shown to improve after intubation simulation training [76]. The team-based intubation safety concept may minimize dETT and other risks of intubation, regardless of the level of training or credentials of the primary intubator.

## Figures and Tables

**Figure 1 children-10-00361-f001:**
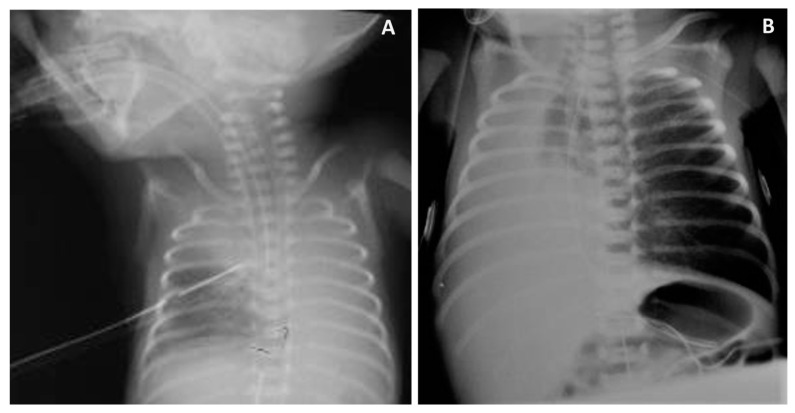
Immediate and delayed sequelae of endobronchial intubation in extremely preterm neonates. (**A**) Intensive delivery room resuscitation and pneumothorax in ELBW newborn. (**B**) Initial right mainstem bronchus intubation with endobronchial mucosal injury in VLBW neonate followed by persistent right lung atelectasis post-extubation.

**Figure 2 children-10-00361-f002:**
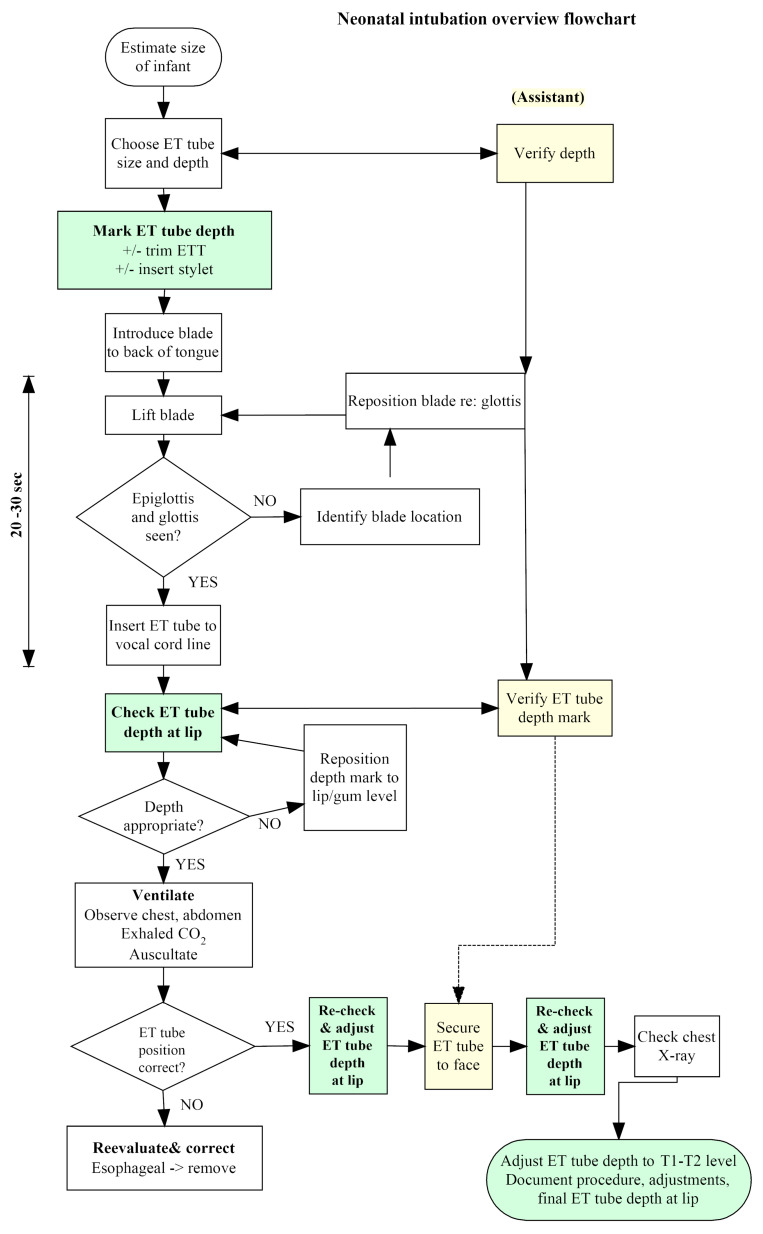
Neonatal intubation flow diagram, emphasizing ETT depth (green rectangles) as a critical control point in intubation safety. Note the role of the intubation assistant in double-checking ETT depth (yellow rectangles) at 3 key stages of the process.

**Figure 3 children-10-00361-f003:**
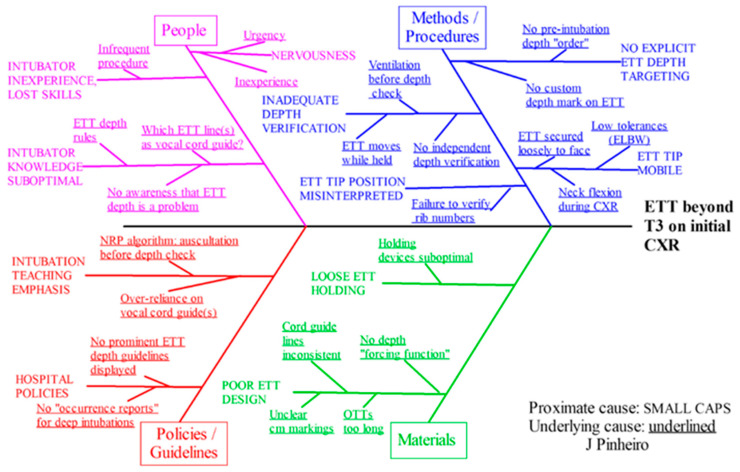
Ishikawa diagram showing factors believed to contribute to deep intubation, categorized by domain.

**Figure 4 children-10-00361-f004:**
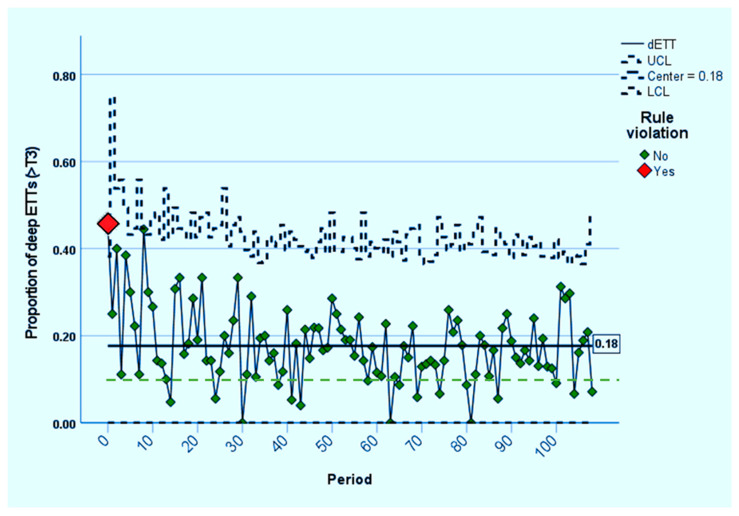
Control chart (p-chart, with 3 sigma upper and lower control limits, UCL, LCL), showing the proportion of dETTs (>T3) at baseline and during the subsequent 108 monthly periods. The only point violating statistical control rules (red diamond symbol) was at baseline (Period 0). Still, the centerline (mean rate) and most monthly points remained above the 10% target (dashed green line).

**Figure 5 children-10-00361-f005:**
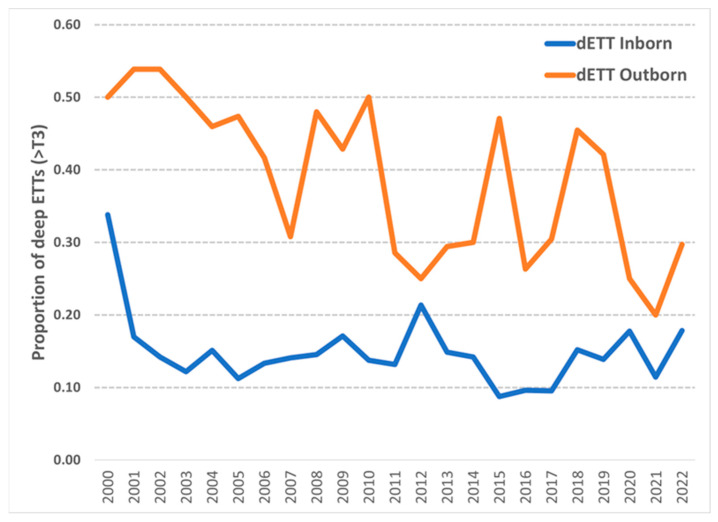
Proportion of dETTs by year, for newborns intubated by AMC staff (“Inborn”) compared to those intubated by staff at referring hospitals (“Outborn”).

## Data Availability

The underlying data are protected for institutional quality improvement purposes, and thus unavailable for use outside the Institution.

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
