# Peer review of "Strategies to Improve Neonatal Intubation Safety by Preventing Endobronchial Placement of the Tracheal Tube—Literature Review and Experience at a Tertiary Center"

_children, 2023, doi:10.3390/children10020361_

Round 1

Reviewer 1 Report

This is an original retrospective analysis of improving avoidance of too deep intubation for neonates over 22 years period. Also the autor did wide range of reviews over causes of too deep intubation in neonates. 

Author Response

We thank the Reviewer for the effort spent in evaluating our complex manuscript, and for the encouraging remarks.

Reviewer 2 Report

This thoughtful and informative paper by Pinheiro et al makes important contributions to our understanding of the strategies to reduce the incidence of deep endotracheal tube placement in neonatal intubations. 

1.     This paper is based on a well-designed Quality improvement project at Albany Medical College NICU. This paper includes a large dataset with 5754 intubations collected over approximately 22 years evaluating the rate of deep ETT placement at baseline and after a series of interventions.

2.      The methodology and results are explained clearly and in detail and the conclusions are supported by the results. 

3.     This paper addresses an important and relatively understudied area of neonatal resuscitation and adds important information to the literature that can help future studies evaluate the incidence of deep ETT placement in neonatal intubation and potentially reduce it.

4.     The discussion was thorough with adequate citation of references. 

5.     My only question to the authors is if you have the data on the adverse consequences mentioned in the paper, including atelectasis, pneumothorax, and other complications that could result from deep placement of the endotracheal tube at baseline and after the various interventions. This would be helpful in evaluating the clinical impact of dETT and of reducing its incidence.

Author Response

This thoughtful and informative paper by Pinheiro et al makes important contributions to our understanding of the strategies to reduce the incidence of deep endotracheal tube placement in neonatal intubations. 

  1. This paper is based on a well-designed Quality improvement project at Albany Medical College NICU. This paper includes a large dataset with 5754 intubations collected over approximately 22 years evaluating the rate of deep ETT placement at baseline and after a series of interventions.
  2. The methodology and results are explained clearly and in detail and the conclusions are supported by the results. 
  3. This paper addresses an important and relatively understudied area of neonatal resuscitation and adds important information to the literature that can help future studies evaluate the incidence of deep ETT placement in neonatal intubation and potentially reduce it.
  4. The discussion was thorough with adequate citation of references. 
  5. My only question to the authors is if you have the data on the adverse consequences mentioned in the paper, including atelectasis, pneumothorax, and other complications that could result from deep placement of the endotracheal tube at baseline and after the various interventions. This would be helpful in evaluating the clinical impact of dETT and of reducing its incidence.

We thank Reviewer 2 for effort spent in evaluating our paper, and for the supportive comments including the recognition that this topic is understudied.

Regarding point #5., we do not have data on immediate or delayed sequelae associated with each intubation. Like the Reviewer, we recognize the usefulness of such information, and we pointed to its absence as a significant limitation, as noted in lines 447-449 (lines 457-458 of the revised version). However, as explained in lines 256 and following (revised version), we did not have the resources to collect the extensive data on the many potential complications, nor did we want to stigmatize the deep ETT events by associating them with morbidities that might or might not be related, in individual instances (e.g., IVH, pneumothorax). 

Reviewer 3 Report

This is an excellent study on a difficult topic, a long term clinical study aimed at improving the technique. The methods and results of the study are easy to understand, but unfortunately the discussion is too verbose and difficult to follow. Please add subheadings to the discussion section, as used in the methods section, to clarify what is being discussed. Figure 4 is more difficult to understand than the other figures, so please explain it in more detail.

Author Response

This is an excellent study on a difficult topic, a long term clinical study aimed at improving the technique. The methods and results of the study are easy to understand, but unfortunately the discussion is too verbose and difficult to follow. Please add subheadings to the discussion section, as used in the methods section, to clarify what is being discussed. Figure 4 is more difficult to understand than the other figures, so please explain it in more detail.

We thank Reviewer 3 for the careful review of this long paper and appreciate the Reviewer’s acknowledgement of the difficulty in this topic. Regarding the suggestions for improvement is the paper:

  1. We recognize that the Discussion is long and difficult to follow. This section contains essentially a summary of a scoping review on potential methods to further decrease endobronchial intubation risk, which could conceivably be expanded into a separate paper. However, each of the methods discussed needed to be mentioned in the context of this paper, as potential future steps, which readers would benefit from considering. So, we labored to fit that contents in the usual format of the Discussion. We thank the Reviewer for suggesting the addition of subheadings, which provides a logical solution to our problem, and will help readers efficiently navigate the various subjects covered in the text, quickly finding those of most interest. Although we trimmed some text in the Discussion, this could not be done to a substantial degree without loss of understanding or without deletion of relevant references. We believe that these changes address the manuscript’s major weakness, which was the density of the discussion.
  2. Regarding Figure 4, we recreated the figure and made some changes in the default output of the SPSS software, to make it clearer to readers. We replaced the standard “nonconforming” with “deep ETTs” in the Y-axis title, and replaced the red circle symbol with a red diamond to be consistent with the auto-generated graph legend. We also added a target line at 10%, to emphasize our aim at the beginning of the project. We clarified this in the Figure 4 legend, along with explaining the meaning of UCL and LCL abbreviations, for readers who may not be fully familiar with control chart terminology.

Round 2

Reviewer 3 Report

Thank you for the appropriate correction as requested. I am impressed that you are able to translate years of clinical research into a paper in this manner.